# An Information-Theoretic Measure for Balance Assessment in Comparative Clinical Studies

**DOI:** 10.3390/e22020218

**Published:** 2020-02-15

**Authors:** Jarrod E. Dalton, William A. Benish, Nikolas I. Krieger

**Affiliations:** 1Department of Quantitative Health Sciences, Cleveland Clinic, Cleveland Clinic Lerner College of Medicine at Case Western Reserve University, 9500 Euclid Avenue, Cleveland, OH 44126, USA; kriegen@ccf.org; 2Department of Internal Medicine, Case Western Reserve University, Cleveland, OH 44106, USA; wab4@cwru.edu

**Keywords:** balance, Jensen–Shannon divergence, observational study, relative entropy, selection bias

## Abstract

Limitations of statistics currently used to assess balance in observation samples include their insensitivity to shape discrepancies and their dependence upon sample size. The Jensen–Shannon divergence (JSD) is an alternative approach to quantifying the lack of balance among treatment groups that does not have these limitations. The JSD is an information-theoretic statistic derived from relative entropy, with three specific advantages relative to using standardized difference scores. First, it is applicable to cases in which the covariate is categorical or continuous. Second, it generalizes to studies in which there are more than two exposure or treatment groups. Third, it is decomposable, allowing for the identification of specific covariate values, treatment groups or combinations thereof that are responsible for any observed imbalance.

## 1. Introduction

The goal of comparative studies is to measure the effect of two or more treatment (or exposure) groups on an outcome. A potential source of bias in these studies is the association between the treatment groups and one or more confounding variables. Randomized clinical trials mitigate this risk through randomization of treatments, resulting in balanced groups with respect to the confounding variables. We say that the relationship between treatment *T* and outcome *O* is confounded by a covariate *C* if *C* is associated with *O* and *T* but is not a consequence of *T* (i.e., not a mediator of the effect of *T* on *O*) [1].

A common strategy for evaluating the potential for confounding in such a study is to identify all covariates that may meet these criteria and evaluate their association with *T*. When treatment groups *T* are balanced on a variable *C*, that is, when *T* and *C* are probabilistically independent, then *C* cannot confound the estimation of the relationship between *T* and *O*.

A variety of techniques are typically employed to assess balance in observational samples, including estimation of simple univariate descriptive statistics, univariate tests of association, and estimation of standardized difference scores (defined as the difference in means between groups divided by a combined estimate of standard deviation). Depending on the situation, however, each of these three approaches may lead to erroneous conclusions. Univariate descriptive statistics may not adequately capture complex distributions (e.g., those with multiple modes) [2]. Tests of association are heavily dependent on sample size, and thus can be as indicative of sample size as they are of imbalance. And standardized difference scores—despite their popularity—are not sensitive to discrepancies in higher order moments (e.g., skewness, kurtosis) and/or multimodalities among continuous distributions.

In this article, we propose the use of an information-theoretic measure known as the Jensen–Shannon divergence (JSD) [3] to assess treatment group balance. The JSD offers several advantages over the aforementioned approaches. First, it is universally defined for binary, multilevel, and continuous distributions (although, in practice, computation for continuous distributions is facilitated by binning the variables into a number of discrete levels), for any number of treatment groups, and for multivariate distributions (i.e., vectorized covariate values $\vec{C}$) across treatment groups. Second, it allows for the identification of specific levels of *C* or *T*—and, moreover, specific combinations of *C* and *T*—that contribute most to imbalances across groups or treatments in relation to others. And third, it is sensitive to high order imbalances (e.g., differences in variability, skewness, bimodality, etc.) in addition to location shifts.

A brief introduction to information theory and the JSD is presented in the next section of this report (Section 2). Properties of the JSD as a measure of covariate imbalance are discussed in Section 3. Examples are presented in Section 4. We conclude with a brief summary (Section 5).

## 2. Information Theory and the JSD

The JSD is an information-theoretic measure of dissimilarity among two or more probability distributions [3]. It is derived from relative entropy (or Kullback–Leibler divergence) [4] and is therefore related to mutual information [5] (pp. 18–21). These measures are fundamentally tied to Shannon’s entropy [6]. The goal of this section is to describe the JSD in intuitive terms, beginning with the definition of entropy.

### 2.1. Entropy

Let *X* be a discrete random variable which takes on values $x_{i}\in \left\{x_{1},\;x_{2},\ldots ,x_{M}\right\}.$ Let the probability distribution of *X* be denoted as f(X). The entropy of *X*, denoted H(X), is a measure of the uncertainty of the outcome of *X* and is defined as:(1)$$H\left(X\right)=E(-\log_{2}f\left(X\right))=-\overset{M}{\underset{i=1}{\sum }}f\left(x_{i}\right)\log_{2}f\left(x_{i}\right).$$

The base of the logarithm is arbitrary. Log base two is often used, giving entropy units of bits (binary digits).

One approach to understanding the concept of entropy is to explore its relationship to the average number of bits (e.g., 0 s and 1 s) required to efficiently encode a sequence of outcomes of the random variable. Consider, for example, the case where the sample space is {A,B,C,D} with corresponding probabilities f(X)={0.25, 0.125, 0.5, 0.125}. With four possible outcomes, it may be tempting to encode a single outcome using two bits, e.g., 00 → *A*, 01 → *B*, 10 → *C*, and 11 → *D*. A more efficient mapping is 0 → *C*, 10 → *A*, 110 → *B*, and 111 → *D*. Since *A*, *B*, *C*, and *D* have probabilities of 0.25, 0.125, 0.5, and 0.125, respectively, and are encoded with 2, 3, 1, and 3 bits, respectively, the expected value of the number of bits required to transmit the outcome of X with this coding scheme is 0.25 × 2 bits + 0.125 × 3 bits + 0.5 × 1 bit + 0.125 × 3 bits = 1.75 bits. 

Shannon demonstrated that H(X) defines a limit beyond which codes cannot be made more efficient. Using either of the above coding schemes allows for the unambiguous encoding of a series of outcomes of X, but the second scheme is optimal in that the expected number of bits required to transmit the outcome of X is H(X) = 1.75 rather than two. To achieve (or to become arbitrarily close) to the efficiency specified by H(X) may require a mapping that associates each code with a sequence of outcomes of X [5] (p. 104). For example, in the case of two possible outcomes *A* and *B*, with respective probabilities 2/3 and 1/3, a code that is more efficient than simply 0 → *A* and 1 → *B* is 0 → *AA*, 10 → *AB*, 110 → *BA*, and 111 → *BB*. The length of the inefficient code required to indicate the outcome is 1 bit, but the average length of the more efficient code, per outcome, is 0.9444 bits (compared to the ideal of H(X) = 0.9183 bits). 

### 2.2. Joint and Conditional Entropy

The joint entropy H(X,Y) of two random variables is a natural extension of the concept of entropy for a single random variable:(2)$$H\left(X,Y\right)=E(-\log_{2}f\left(X,Y\right))=-\overset{M}{\underset{i=1}{\sum }}\overset{N}{\underset{j=1}{\sum }}f\left(x_{i},y_{j}\right)\log_{2}f\left(x_{i},y_{j}\right).$$

Similar to that described above for a single random varible, the joint entropy defines the lower limit of the average number of bits required to encode the observations from the joint distribution.

Conditional entropy, denoted H(X|Y), is a measure of residual uncertainty in X, given the observation of some other random variable Y. It is defined as:(3)$$H\left(X|Y\right)=E(-\log_{2}f(X|Y))=-\overset{M}{\underset{i=1}{\sum }}\overset{N}{\underset{j=1}{\sum }}f\left(x_{i},y_{j}\right)\;\log_{2}f(x_{i}|y_{j}).$$

Conditional entropy is also equal to the difference between joint and marginal entropies, i.e., H(X|Y)=H(X,Y)−H(Y). In this sense, conditional entropy represents the number of bits needed to encode X after the value of Y is observed. Joint and conditional entropy naturally extend to distributions that are defined across three or more random variables (we omit these equations for the purposes of this discussion).

### 2.3. Mutual Information

The mutual information between the random variables X and Y, denoted I(X;Y), is the expected value of the amount of information that knowledge of the outcome of *Y* provides about the outcome of *X*. Mutual information is symmetric with respect to X and Y, and is a function of both the variables’ marginal entropies and their joint entropy:(4)$$\begin{array}{ll} I\left(X;Y\right) & =H\left(X\right)+H\left(Y\right)-H\left(X,Y\right)\\ & =H\left(X\right)-H\left(X|Y\right)\\ & =H\left(Y\right)-H\left(Y|X\right)\\ & =I\left(Y;X\right). \end{array},$$

### 2.4. Relative Entropy

Relative entropy is an information-theoretic measure expressing the divergence from a given probability distribution f(X) to a reference (or target) distribution g(X). It is defined as
(5)$$D\left(g\left(X\right)∥f\left(X\right)\right)=E_{g\left(X\right)}\left[-\log_{2}\frac{f\left(X\right)}{g\left(X\right)}\right]=\overset{M}{\underset{i=1}{\sum }}g\left(x_{i}\right)\log_{2}\frac{g\left(x_{i}\right)}{f\left(x_{i}\right)}.$$

The relative entropy is interpreted as the number of bits required to “correct” the probabilities in the distribution f so that they match those of the reference distribution g (under an optimal coding scheme) [5] (p. 18).

Since the expectation in Equation (5) is taken with respect to the target distribution g(X), the relative entropy function is asymmetric, i.e., it is not necessarily the case that D(g(X)∥f(X))=D(f(X) ∥g(X)). Given this asymmetry, it is not a suitable candidate for a measure of covariate balance among groups: the divergence between two groups would depend upon which group is taken to be the Reference group. Jeffrey’s divergence (*J*) is a symmetric version of relative entropy, defined as J(g(x);f(x))=D(g(x)||f(x))+D(f(x)||g(x)) [7]. One reason why it is not a suitable candidate for the task of assessing covariate balance among groups is that there may be more than two groups.

### 2.5. Jensen–Shannon Divergence (JSD)

The JSD is a modified version of relative entropy, that addresses the asymmetry problem described above by expressing divergences with respect to a common distribution $\tilde{f}\left(X\right)$. Assume that there are *N* distributions of X: $f_{1}\left(X\right),\;f_{2}\left(X\right),\ldots ,f_{N}\left(X\right).$ The common distribution is taken as the (unweighted) mean of the component densities:(6)$$\tilde{f}\left(x\right)=\frac{1}{N}\overset{N}{\underset{k=1}{\sum }}f_{k}\left(x\right).$$

The JSD of the set of distributions $f_{k}\left(X\right)$ is defined as the average relative entropy from the common distribution $\tilde{f}\left(X\right)$ to the specific distributions $f_{k}\left(X\right):$(7)$$\mathrm{JSD}=\frac{1}{N}\overset{N}{\underset{k=1}{\sum }}D\left(f_{k}\left(X\right)∥\tilde{f}\left(X\right)\right).$$

### 2.6. The JSD of Covariate Distributions Across Treatment Groups

Equations (6) and (7) can be modified to calculate the JSD for a set of *N* treatment groups. We replace the continuous random variable X with the discrete covariate random variable C. Similary, we replace the probability density function f with the probability mass function p. Assuming that *C* can assume *M* values, we have, for i=1, ⋯, M:(8)$$\tilde{p}\left(c_{i}\right)=\frac{1}{N}\overset{N}{\underset{k=1}{\sum }}p_{k}(c_{i}),$$
and
(9)$$\mathrm{JSD}=\frac{1}{N}\overset{N}{\underset{k=1}{\sum }}D\left(p_{k}\left(C\right)∥\tilde{p}\left(C\right)\right)=\frac{1}{N}\overset{N}{\underset{k=1}{\sum }}\overset{M}{\underset{i=1}{\sum }}p_{k}\left(c_{i}\right)\log_{2}\left(\frac{p_{k}\left(c_{i}\right)}{\frac{1}{N}\sum_{k=1}^{N}p_{k}(c_{i})}\right).$$

## 3. Properties of the JSD

The JSD is non-negative and is equal to zero when the covariate distributions are identical for all treatment groups. It is interpreted as the average relative entropy from the common covariate distribution, $\tilde{f}\left(C\right)$, to the group-specific distributions. As noted in the Introduction, the JSD can be applied to binary random variables, categorical random variables, or continuous random variables.

Being defined additively in terms of units of information, the JSD is decomposable. One may calculate the JSD across all the treatment groups or determine the contribution of a subset of groups to the overall JSD. Similarly, specific levels of the covariate(s) of interest may be examined to identify regions of the covariate space exhibiting the greatest degree of imbalance across groups. Furthermore, contributions of individual treatment/covariate combinations to the overall JSD can be studied and compared. The decomposability of the JSD is illustrated in Section 4.

As a function of the densities themselves (and not their moments), the JSD allows for the evaluation of balance in a manner that does not assume that continuous densities belong to any particular family of distributions. It is sensitive to shape discrepancies among groups. In contrast, the standardized difference score converges to zero (with increasing group sample sizes) whenever the means of the two samples are equal (see Figure 1).

In practice, computation of the JSD using observational data can be difficult for continuous densities, especially mixture distributions [2]. Our approach relies on the binning of continuous variables (as is done with histograms). When small numbers of categories are used, this simplification can mask subtle features of group-specific probability densities. A further limitation of the JSD is that density estimates for categorical variables are increasingly variable among small samples.

## 4. Applications

Table 1 summarizes findings from 93,583 outpatients in the Cleveland Clinic Health System who had a lipid panel drawn between 2007 and 2010 (first visit meeting these criteria). The patients are partitioned into three treatment groups: Disadvantaged (age < 80 years and living in a census tract that is in the top 25% of all tracts in the United States with respect to the Area Deprivation Index [8]), Elderly (not living in a disadvantaged neighborhood per the above definition but aged 80 or older), and Reference (neither disadvantaged nor elderly). The covariate is baseline diabetes state defined by blood sugars < 109 mg/dL, 109–125 mg/dL, and > 125 mg/dL. A stand-alone R package for implementing the JSD computations illustrated in this section is provided at http://github.com/jarrod-dalton/jsd, and the code used for this section is given in the Appendix A.

Table 2 presents the probability distributions of glucose levels within each treatment group. The average of these distributions, i.e., the common distribution, $\tilde{f}\left(C\right),$ is shown in the final column.

Table 3 presents contributions of individual cells, the three treatment groups, and the three covariate groups to the overall JSD, which is 0.0144 bits. This is the average of the relative entropies from the common distribution to the treatment group-specific distributions. Given three treatment groups, the maximum possible JSD is $\log_{2}\left(3\right)=1.5850$ bits.

The Reference group is the largest treatment group contributor to the JSD, and the Glucose > 125 category is the largest covariate group contributor to the JSD. Moreover, by considering the absolute values of the individual cell components, we conclude that the largest contributor to the JSD is from individuals in the Reference group with serum glucose values less than 109 mg/dL.

A problem with using any method to quantify covariate imbalance among treatment groups is that there is no obvious point that defines an acceptable amount of imbalance [9]. For the current example, the JSD value of 0.0144 bits is small relative to its maximal possible value of 1.5850 bits, but it is clear from Table 2 that individuals in the Reference group tend to have lower blood sugars than individuals in the other two treatment groups. An important factor in deciding what constitutes acceptable balance is the potential of the covariate to affect the outcome [10].

In order to further examine differences between the JSD and standardized difference scores, we consider the case in which there are two treatment groups with normally distributed covariates. Figure 2 plots the JSD as a function of the standardized difference score, when the standard deviation of one of the two distributions is one and the standard deviation of the other distribution is either one (plotted in black), two (plotted in blue), or three (plotted in red). Since there are two treatment groups, the JSD curves asymptote at one bit (since $\log_{2}\left(2\right)=1).$ The standardized difference score curves, on the other hand, are unbounded in the positive direction. As expected, both the JSDs and the standard difference scores increase as the two distributions diverge. The plot also illustrates the point made in Section 3 that the JSD, but not the standardized difference score, is sensitive to differences between the standard deviations of the two distributions when the means of two distributions are identical.

## 5. Summary

We propose that the JSD be used to assess treatment group balance on known potential confounding variables in comparative clinical studies. This information-theoretic measure is equal to the average relative entropy between the covariate distributions for each treatment group and a common distribution, defined as the average of the individual distributions. Advantages of the JSD over alternative measures of treatment group balance include its sensitivity to the shape of distributions and its insensitivity to sample size. The JSD is applicable to both categorical and continuous random variables. Moreover, the JSD is decomposable, allowing for comparisons among specific levels of covariates of interest.

## Figures and Tables

**Figure 1 entropy-22-00218-f001:**
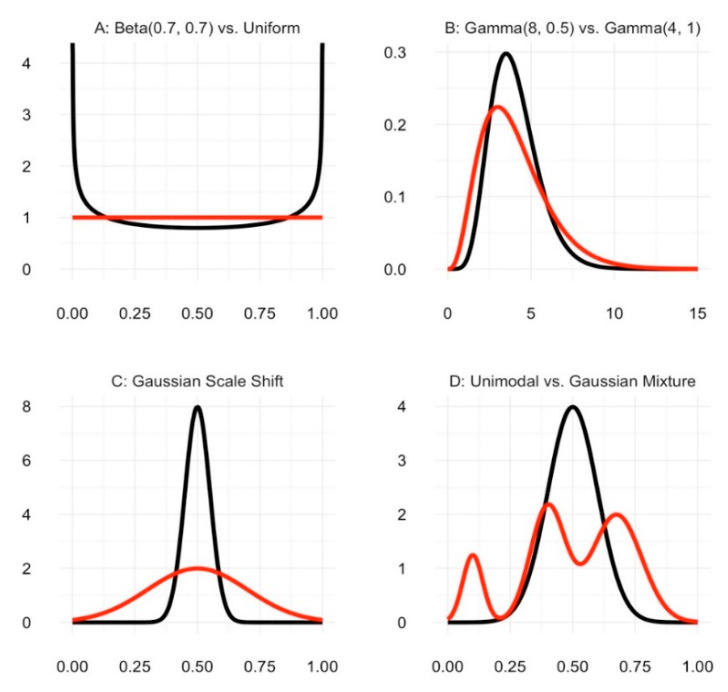
Four pairs of continuous distributions, each of which has a standardized difference score equal to zero.

**Figure 2 entropy-22-00218-f002:**
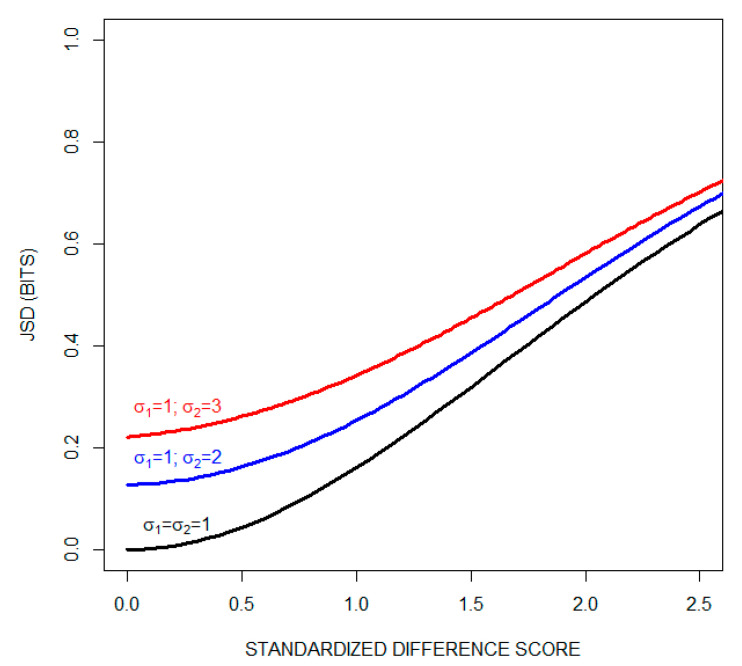
The JSD as a function of the standardized difference score when there are two treatment groups with normally distributed covariates. Three cases are shown: the standard deviation of one of the two distributions is set equal to one, while the standard deviation of the second distribution is set to equal either one (black curve), two (blue curve), or three (red curve).

**Table 1 entropy-22-00218-t001:** Number of individuals in three treatment groups (Disadvantaged, Elderly, Reference) and three covariate groups (defined by blood sugar ranges).

Glucose	Disadvantaged	Elderly	Reference
<109	7191	3637	64,265
109–125	1025	835	7298
>125	1715	685	6932

**Table 2 entropy-22-00218-t002:** Probability distributions of glucose levels within each treatment group (Disadvantaged, Elderly, Reference). The common distribution, $\tilde{f}\left(C\right),\;$ is shown in the final column.

Glucose	Disadvantaged	Elderly	Reference	$\tilde{f}\left(C\right)$
<109	0.724	0.705	0.819	0.749
109–125	0.103	0.162	0.093	0.119
>125	0.173	0.133	0.088	0.131

**Table 3 entropy-22-00218-t003:** Contributions of individual cells, treatment groups, and levels of the covariate to the overall JSD (in units of bits).

Glucose	Disadvantaged	Elderly	Reference	Total
<109	−0.0119	−0.0206	0.0349	0.0023
109–125	−0.0072	0.0237	−0.0112	0.0053
>125	0.0228	0.0008	−0.0168	0.0067
Total	0.0036	0.0039	0.0068	0.0144 *****

* Note: row/column sums do not equal 0.0144 due to rounding error.

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
