# Peer review of "An Information-Theoretic Measure for Balance Assessment in Comparative Clinical Studies"

_entropy, 2020, doi:10.3390/e22020218_

Round 1
Reviewer 1 Report
Review letter of "The Jensen-Shannon Divergence as a Measure of Covariate Balance for Observational Samples" by Dalton & Benish (2019)
The authors developed an interesting paper related to the use of Jensen-Shannon Divergence (JSD) in clinical observations. They highlighted the benefits of JSD on some particular kind of distributions, and then, in an application related to patients with glucose concentrations.
I recommend to authors to address the following comments, in order to following the review for further publication process. Also, consider some reference to help to improve your manuscript.
Mayor/Moderate comments:
In all parts, for JSD, use "Distance" word, not "Divergence". This is, because JSD accomplish the triangular inequality, in addition, to symmetry property. L 116: However, the Jeffreys’ (J) divergence [Crooks 1967] is used as a symmetric version of this divergence: J(g(X)||f(y)) = D(g(X)||f(Y) ) + D(f(Y)||g(X)). L 141-143: Take into account that the exact expression of JSD for mixture of continous densities are not easy to obtain. In these cases, Shanon or Rényi entropies can be obtained (Contreras-Reyes and Cortés, 2016). This disadvantage of JSD could be commented at the end of Section 3. Figure 2: Please, use a better format for this figure. Also, it could be centered. Save the figure in pdf format in R software.
Minor comments:
L 11 & 16: "Jenson" <-> "Jensen". L 34: ...modes [Contreras-Reyes and Cortés, 2016]). L 40 & 53: JSD [2]. L 55-56: This sentence could be moved after Eq. 1. L 89: "entropy". L 106: "Relative entropy or or Kullback-Leibler divergence [3]". L 133-134: Repeated sentence (in lines 41-44), please delete it here. L 152: "Applications". L 166: "... bits, as is presented in Table 3". L 252: Delete this line (informal).
References:
1. Crooks, G. E. (1967). Inequalities between the Jenson–Shannon and Jeffreys divergences. Studia Sci. Math. Hungar, 2, 299-318.
2. Contreras-Reyes, J.E., Cortés, D.D. (2016). Bounds on Rényi and Shannon Entropies for Finite Mixtures of Multivariate Skew-Normal Distributions: Application to Swordfish (Xiphias gladius Linnaeus). Entropy, 18(11), 382.
Reviewer 2 Report
The modification of divergence entropy is shown to avoid the unsymmetry of the original definion.
The comments:
eq 3 - is it correct ? - the sign of dependence - isn it missing on right site of the equation ? The step between eq 5 to eq 7 shall be more clariffied the step between theory oan application is to short - Explanation how the values shown in Tables shall be goven in a more easy way to make possible application by non-specialistsAuthor Response
Please see the attachment.

Round 2
Reviewer 1 Report
Thanks for addressing all of my comments in well form. I do not have further observations.